# Structure-Based Pharmacophore Modeling, Virtual Screening, Molecular Docking and Biological Evaluation for Identification of Potential Poly (ADP-Ribose) Polymerase-1 (PARP-1) Inhibitors

**DOI:** 10.3390/molecules24234258

**Published:** 2019-11-22

**Authors:** Yunjiang Zhou, Shi Tang, Tingting Chen, Miao-Miao Niu

**Affiliations:** Department of Pharmaceutical Analysis, State Key Laboratory of Natural Medicines, School of Basic Medicine and Clinical Pharmacy, China Pharmaceutical University, Nanjing 210009, China; zyjcqmu@163.com (Y.Z.); ts2016ts@126.com (S.T.); jyjq23@163.com (T.C.)

**Keywords:** PARP-1, virtual screening, pharmacophore modeling, molecular docking

## Abstract

Poly (ADP-ribose) polymerase-1 (PARP-1) plays critical roles in many biological processes and is considered as a potential target for anticancer therapy. Although some PARP-1 inhibitors have been reported, their clinical application in cancer therapy is limited by some shortcomings such as weak affinity, low selectivity and adverse side effects. To identify highly potent and selective PARP-1 inhibitors, an integrated protocol that combines pharmacophore mapping, virtual screening and molecular docking was constructed. It was then used as a screening query to identify potent leads with unknown scaffolds from an in-house database. Finally, four retrieved compounds were selected for biological evaluation. Biological testing indicated that the four compounds showed strong inhibitory activities on the PARP-1 (*IC_50_* < 0.2 μM). MTT assay confirmed that compounds **1**–**4** inhibited the growth of human lung cancer A549 cells in a dose-dependent manner. The obtained compounds from this study may be potential leads for PARP-1 inhibition in the treatment of cancer.

## 1. Introduction

Poly (ADP-ribose) polymerases (PARPs) are abundant and ubiquitous nuclear proteins encoded by 18 different genes and involved in many basic processes, including control of cell death, DNA repair and transcriptional regulation [1,2,3]. Among the PARP family, PARP-1 is the most typical characteristic member of the PARP family and has the following structural regions: DNA-binding, self-modifying and catalytic [4,5]. The catalytic domain is responsible for transferring the ADP-ribose portion from nicotinamide adenine dinucleotide (NAD^+^) to the substrate protein [6,7,8]. When triggered by DNA damage, PARP-1 can cause a drop in the level of NAD^+^ and the consumption of intracellular ATP [7,8]. This depletion of ATP leads to cell dysfunction and necrotic cell death [7,8]. Many studies with PARP-1 knockout mice have demonstrated PARP-1 plays crucial roles in DNA cell repair and survival [8,9,10]. Therefore, inhibition of PARP-1 is a promising therapeutic target for drug development in the context of various forms of ischemia, inflammation and cancer therapy [11].

Recently, the therapeutic application of PARP-1 inhibitors has received considerable attention because of its potential in treating cancer, inflammatory diseases, neurodegenerative diseases and several other diseases [11,12]. So far, most of the PARP-1 inhibitors in clinical development are structural analogues of NAD^+^ [5,13]. These reagents are designed to compete with NAD^+^ at the enzyme activity site and bind to the catalytic domain of the enzyme to inhibit self-modification [13,14]. Although these works reflect the exciting and important development of the PARP-1 inhibitors, there are still many major obstacles such as water insolubility, weak affinity, low selectivity [13,14], high toxicity [15], resistance to high expression of genes or proteins [16,17,18,19] and adverse reactions to the blood system [18,19]. For example, the new drug olaparib has adverse reactions such as fatigue, nausea and vomiting, and drug resistance [19,20,21,22,23]. Niraparib shows common adverse reactions in the clinic, including thrombocytopenia, anemia, nausea, vomiting hypertension and other toxic reactions [24]. Therefore, there is a pressing need to develop novel, selective and high-affinity PARP-1 inhibitors.

Pharmacophore-based virtual screening is nowadays a mature technology, very well accepted in the medicinal chemistry laboratory [25]. Compared with molecular docking, it can more accurately and rapidly identify novel and potent inhibitors of other targets by common feature-based alignment [26]. In previous works, such pharmacophore-based procedures were successfully used to detect some positive hits in other targets such as histone methyltransferase 7, soluble epoxide hydrolase and enhancer of zeste homolog 2 [27,28,29]. In this work, a pharmacophore model was constructed based on the X-ray crystallographic structures of the PARP-1 with a high resolution. In silico screening of a virtual compound database using the combined pharmacophore modeling and molecular docking studies led to the successful identification of several PARP-1 inhibitors, of which structures are different from existing PARP inhibitors such as olaparib and niraparib (Appendix A). This study has shown that it is feasible to utilize in silico screening approaches to rationally design PARP-1 inhibitors with possible utility in the treatment of cancer. The present study is expected to provide an effective guide for methodical development of potent PARP-1 inhibitors.

## 2. Results and Discussion

### 2.1. Pharmacophore Modeling

Computer-aided drug design (CADD) plays a vital role in drug discovery and has become an important tool in the pharmaceutical industry. To obtain all available chemical information on the inhibitor binding of the PARP-1, a structure-based pharmacophore model was constructed based on the three X-ray crystallographic structures of the PARP-1 catalytic domain with a high resolution of less than 2.5 Å (Table 1). The generated structure-based model included four features (Figure 1A): one hydrogen bond donor feature (F1: Don), one hydrogen bond acceptor feature (F2: Acc), one aromatic feature (F3: Aro) and one hydrophobic feature (F4: Hyd). To provide spatial limitation of the ligand size and shape, a steric constriction was added to the structure-based pharmacophore model (Figure 1B). The features of the model were found to directly interact with key amino acids, such as Tyr907, Gly863, Ser904, Lys903, Ala898, Phe897, Tyr896, Tyr889, Leu765, Asp766 and Val762, which play a major role in PARP-1 inhibition activity. Therefore, these features can be considered as important chemical features in discovering novel PARP-1 inhibitors.

### 2.2. Validation and Database Screening

To validate the reliability of the structure-based pharmacophore model, the model was employed to screen the testing set database. The testing set included 20 known PARP-1 inhibitors with experimental activity and 1480 inactive molecules [30,31,32,33]. To assess the discriminating ability of the model, the pharmacophore model was used as a 3D structural search query to perform a virtual database searching. Some valuable parameters such as total hits (*Ht*), active hits (*Ha*), enrichment factor (*E*) and goodness of hit score (*GH*) were calculated (Table 2). When the *GH* score is higher than 0.7, the model is very good. It was observed to be 0.8 for the model, indicating that the pharmacophore model showed a good ability to distinguish the active molecules from the inactive ones.

The flowchart of virtual screening used in this study is shown in Figure 2. To confirm the discriminatory ability of the generated pharmacophore model, the pharmacophore model was used as a filtrating tool in virtual screening to identify potential compounds from an in-house database containing 35,000 compounds. Based on a RMSD value less than 0.5 Å, 41 selected compounds were docked into the PARP-1 active site. The docking scores between PARP-1 and 41 compounds were calculated by dG docking scoring function of MOE. Considering a cutoff to classify compounds as either active or inactive, we used a −12 kcal mol^−1^ cutoff in docking score to prune the hit list. Finally, only 4 out of 41 compounds (compounds **1**–**4**) below −12 kcal mol^−1^ were used for further biological testing (Figure 3). A good pharmacophore mapping with 4 compounds on the model is shown in Figure 4.

### 2.3. Biological Validation

To test the binding ability of the four compounds to PARP-1, a PARP-1 inhibition assay was performed. The four selected compounds exhibited stronger inhibition activities towards PARP-1 than the control NU1025 (Table 3 and Appendix A). Among the four compounds, compounds **1** and **4** showed the best inhibitory activity. In addition, the selectivity of compounds **1** and **4** to PARP-1 were higher than that to PARP-2 and PARP-3 (Table 4). To further evaluate the anticancer activity of the four compounds, an MTT assay was used to test their inhibition effects on the proliferation of human lung cancer A549 cells. As shown in Figure 5, compounds **1**–**4** inhibited the growth of A549 cells in a dose-dependent manner. Moreover, compounds **1** and **4** had stronger anti-proliferation effects than compounds **2** and **3**. The above findings suggest that compounds **1** and **4** may be potential leads for PARP-1 inhibition in the treatment of cancer.

To further predict a reasonable binding mode between each compound and PARP-1, the four compounds were docked into the active site of PARP-1. The docking results suggested that there were two major interactions between the four compounds and the PARP-1 active site (Figure 6 and Appendix A): (1) the amide group of each compound formed three hydrogen-bonding interactions with Ser904 and Gly863 that are indispensable for the ligand binding of the PARP-1 [32,33]; (2) these compounds were engaged in hydrophobic interactions with some amino acids, including Tyr907, Lys903, Ala898, Phe897 and Tyr896 [32,33,34,35]. This analysis of binding modes can be helpful for the understanding of a possible mechanism underlying inhibitor selectivity of PARP-1.

## 3. Materials and Methods

### 3.1. Pharmacophore Model Generation and Validation

Three X-ray crystallographic structures of the PARP-1 with a resolution of less than 2.5 Å were available for download from the Protein Data Bank (PDB) database. Hydrogens of these structures were added, their gasteiger partial charges were computed and their energy minimization was performed using the Merck molecular force field 94× (MMFF94×) method as implemented in Molecular Operating Environment (MOE) (Chemical Computing Group Inc, Montreal, Quebec, Canada) [35]. Based on these prepared structures, the pharmacophore protocol of MOE has been used to construct the most representative features of the PARP-1 active site, which are indicated as spheres that represent the essential interaction points with the key residues on ligand binding of the PARP-1. 

A testing set database containing 20 active compounds was used to evaluate the discriminative ability of the pharmacophore model in distinguishing active compounds from the inactive compounds. The database screening was performed using pharmacophore search protocol available in MOE. The Güner–Henry (*GH*) test score is applied to quantify model selectivity [2,36]. The hit lists were analyzed based on the following formula: $$GH=\left(\frac{Ha\left(3A+Ht\right)}{4HtA}\right)\left(1-\frac{\left(Ht-Ha\right)}{\left(D-A\right)}\right)$$
where *D* is the number of molecules in the database, *A* is the number of active molecules in the database, *Ht* is the number of hits retrieved, *Ha* is the number of actives in the hits list, *E* is the enrichment of the concentration of actives by the model relative to random screening without a pharmacophore approach, and *GH* is the Gunner–Henry score [2,36]. The *GH* score ranges from 0 to 1, which indicates a null model and an ideal model.

### 3.2. Virtual Screening

An in-house database containing the approximately two-dimensional (2D) 35,000 compounds has been used for virtual screening because of their structural diversities [36]. Before virtual screening, the conformation import protocol available in MOE is used to convert and minimize the structures of the compounds using the MMFF94× force field when moving from 2D to 3D structures. In the process, multiple conformations per compound were generated and minimized, the hydrogens are added and partial charges computed. Then, we have used Lipinski’s rule to identify compounds from the in-house database, owing to unique structural characteristics of the PARP-1 catalytic domain. Afterward, the pharmacophore search protocol of MOE was used to screen drug-like hits that match the pharmacophore model. Hit compounds can be ranked according to the RMSD values, which is the degree of consistency with the pharmacophore model [37]. To decrease the number of hits, we used 0.5 Å of the maximum RMSD value to prune the hit list.

### 3.3. Structure-Based Molecular Docking

The MOE program was used to perform various steps involved in docking simulation. Protein crystal structure of PARP-1 (PDB ID: 6I8M) was downloaded from Protein Data Bank. The errors presented in the crystal structure of PARP-1, including missing atom names, missing loops, steric clashes and picking alternate conformations, were corrected by the structure preparation protocol available in MOE. Hydrogens were added, partial charges were computed and energy minimization was performed using MMFF94× force field (gradient: 0.05). Molecular docking calculations were done using triangle matcher algorithm and the docking score between PARP-1 and each ligand was calculated by dG docking scoring function of MOE [37,38].

### 3.4. In Vitro PARP Inhibition Assay

Purified recombinant human PARPs from Trevigan (Gaithersburg, MD, USA) was used to determine the IC_50_ values of a PARP inhibitor. The PARP enzyme assay was set up on ice in a volume of 100 μL consisting of 50 mM Tris–HCl (pH 8.0), 2 mM MgCl_2_, 30 μg/mL of DNase activated herring sperm DNA (Sigma, MO, USA), 30 μM [^3^H]nicotinamide adenine dinucleotide (67 mCi/mmol), 75 μg/mL PARP enzyme and various concentrations of the compounds to be tested. The reaction was initiated by incubating the mixture at 25 °C. After 15 min of incubation, the reaction was terminated by adding 500 μL of ice cold 20% (*w*/*v*) trichloroacetic acid. The formed precipitate was transferred onto a glass fiber filter (Packard Unifilter–GF/B) and washed three times with ethanol. After the filter is dried, the radioactivity is determined by scintillation counting.

### 3.5. MTT Assay

A549 cells were seeded in a 96-well culture plate and allowed to grow overnight. Then, cells were exposed to different concentrations of compounds **1**–**4** and incubated at 37 °C for 48 h. After that, an MTT stock solution (0.5 mg/mL) was added into each well and the plate was incubated for 4 h. The 150 µL of DMSO was used for fixing the MTT-treated cells and the absorbance of each sample was recorded at 490 nm with a Microplate Spectrophotometer.

## 4. Conclusions

In summary, an integrated protocol including pharmacophore modeling and molecular docking studies has successfully been developed. The applied virtual screening protocol led to the identification of four hit compounds. Biological testing results suggest that these compounds have a strong inhibitory effect on the PARP-1 and possess significant anti-proliferation effects on human lung cancer cells. It could be expected that compounds **1** and **4**, the most significant PARP-1 inhibitors, can be explored for the further development of new and more potent inhibitors of PARP-1. Structural optimization for compounds **1** and **4** with respect to PARP-1 inhibition is under way. In addition, these results demonstrate that the screening protocol shows great potential in identifying potent PARP-1 inhibitors. This integrated protocol provides guidelines for screening in small-molecule databases or collections of known inhibitors, and probably can be used for other PARP family member in the future. We are presently using the protocol as a 3D query for the identification of novel potential PARP-1 inhibitors in large 3D databases of molecules.

## Figures and Tables

**Figure 1 molecules-24-04258-f001:**
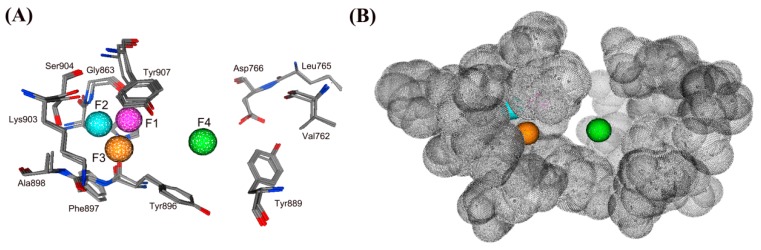
(**A**) The generated pharmacophore model in the binding site of PARP-1 (PDB ID: 6I8M). Pharmacophore features are color-coded: purple, one hydrogen bond donor feature (F1: Don); cyan, one hydrogen bond acceptor feature (F2: Acc); orange, one aromatic feature (F3: Aro); green, one hydrophobic feature (F4: Hyd). Active site residues (dark gray) are shown in line form. Residues are annotated with their three-letter amino acid code. (**B**) A steric constriction (dark gray) was added to the pharmacophore model.

**Figure 2 molecules-24-04258-f002:**
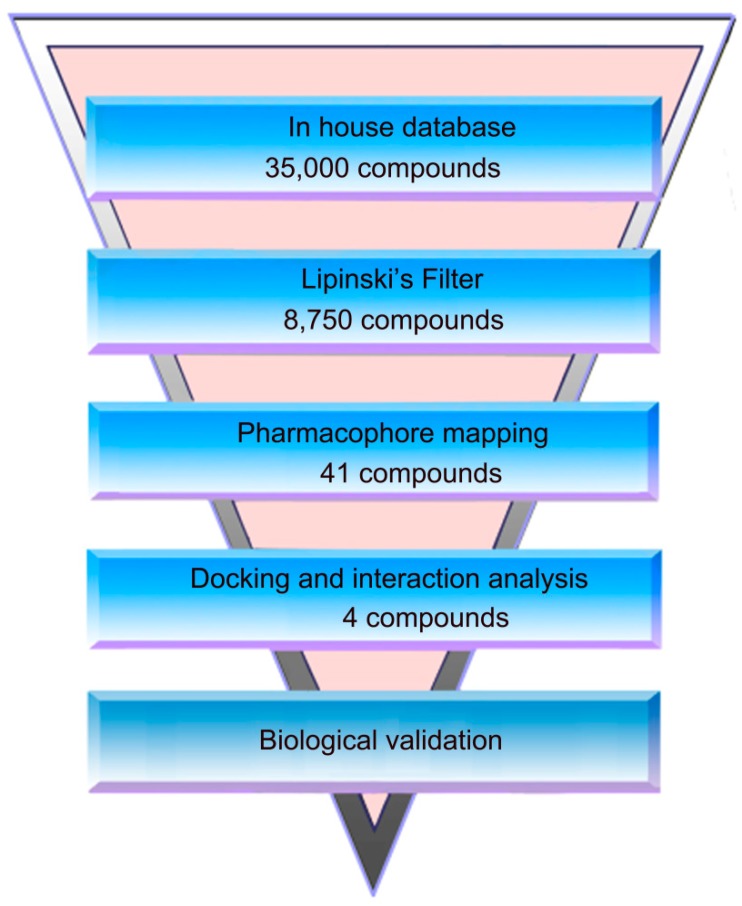
A workflow overview of pharmacophore modeling, selection of compound and biological testing.

**Figure 3 molecules-24-04258-f003:**
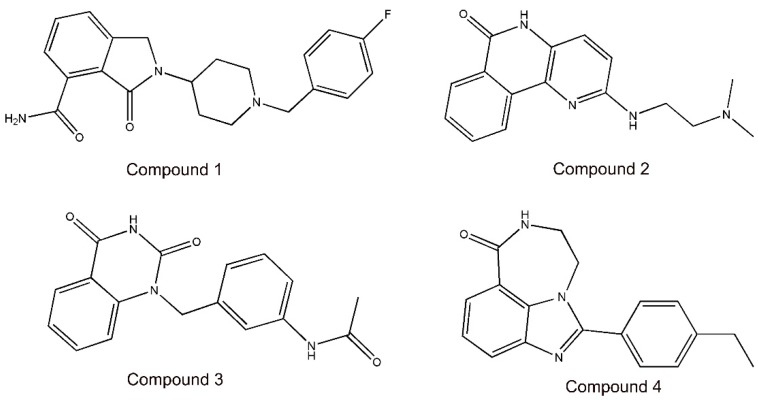
The chemical structures of four selected compounds (**1**–**4**).

**Figure 4 molecules-24-04258-f004:**
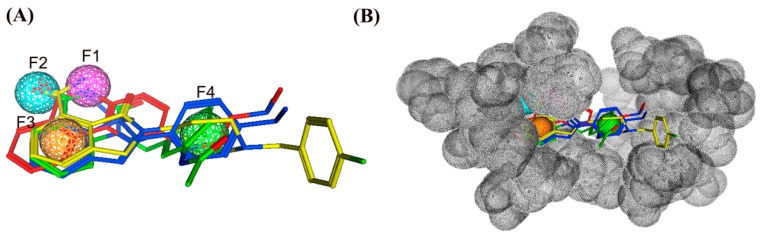
(**A**) The pharmacophore mapping of four selected compounds on the model. Pharmacophore features are color-coded: purple, one hydrogen bond donor feature (F1: Don); cyan, one hydrogen bond acceptor feature (F2: Acc); orange, one aromatic feature (F3: Aro); green, one hydrophobic feature (F4: Hyd). (**B**) A steric constriction (dark gray) was added to the pharmacophore model.

**Figure 5 molecules-24-04258-f005:**
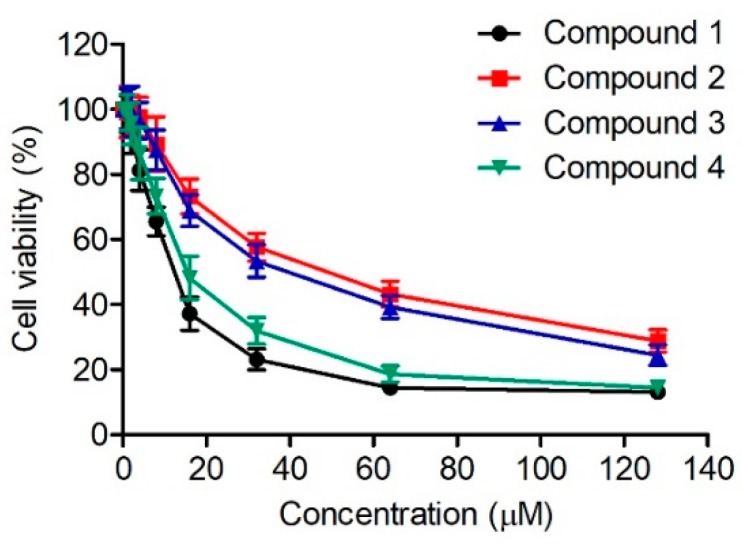
Inhibitory effects of four compounds on the proliferation of A549 cells. The results are representative of three independent experiments and are expressed as mean ± SD.

**Figure 6 molecules-24-04258-f006:**
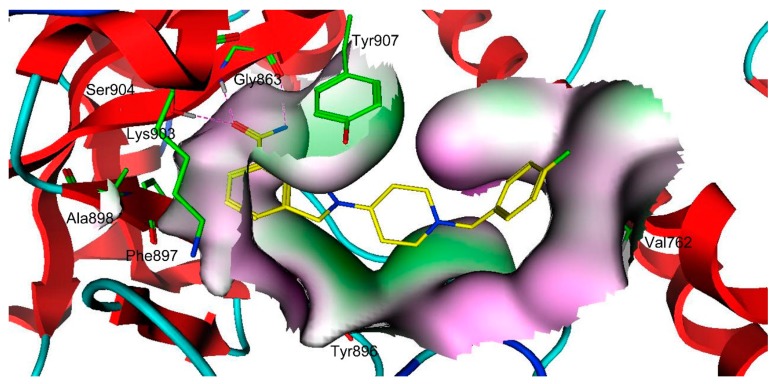
The three-dimensional (3D) ligand–protein interaction diagram for the binding site of PARP-1 with compound **1**. The active site residues are shown in green stick form. The hydrogen-bond network with protein residues is represented in red dotted lines. Compound **1** is color-coded by yellow.

**Table 1 molecules-24-04258-t001:** Basic information of receptor–ligand complexes of the PARP-1 catalytic domain from the PDB database.

PDB_ID	Resolution (Å)	Ligand_ID
4ZZZ	1.9	FSU
5WS1	1.9	7U9
6I8M	2.1	H7W

**Table 2 molecules-24-04258-t002:** Pharmacophore model validation using *GH* score method.

Serial No.	Parameter	Pharmacophore Model
1	Total molecules in database (*D*)	1500
2	Total number of actives in database (*A*)	20
3	Total hits (*Ht*)	23
4	Active hits (*Ha*)	18
5	% Yield of actives ((*Ha*/*Ht*) × 100)	78
6	% Ratio of actives ((*Ha*/*A*) × 100)	90
7	Enrichment factor (*E*) ((*Ha* × *D*)/(*Ht* × *A*))	59
8	False negatives (*A* − *Ha*)	2
9	False positives (Ht − Ha)	5
10	Goodness of hit score (*GH*)	0.8

**Table 3 molecules-24-04258-t003:** Hit compounds selected from an in-house database.

Compounds	RMSD [Å] ^b^	Docking Score [kcal·mol^−1^] ^c^	*IC_50_* [μM]
**1**	0.1596	−13.9	0.031 ± 0.005
**2**	0.3793	−12.1	0.19 ± 0.026
**3**	0.3741	−12.4	0.16 ± 0.013
**4**	0.2363	−12.9	0.065 ± 0.008
NU1025 ^a^	>1	−7.6	0.4 ± 0.035

^a^ Reference 37. ^b^ The root of the mean square distance between the query features and their matching ligand annotation points; ^c^ binding free energy between PARP-1 and a ligand (lower values indicate better binding affinity). The results are representative of three independent experiments and are expressed as mean ± SD.

**Table 4 molecules-24-04258-t004:** Selectivity of compounds **1** and **4** to PARPs.

Compounds	PARP-1 [μM]	PARP-2 [μM]	PARP-3 [μM]	PARP-2/PARP-1	PARP-3/PARP-1
**1**	0.031 ± 0.005	>35	>100	>1000	>3000
**4**	0.065 ± 0.008	>35	>100	>1000	>3000

The results are representative of three independent experiments and are expressed as mean ± SD.

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
