# Peer review of "Structure-Based Pharmacophore Modeling, Virtual Screening, Molecular Docking and Biological Evaluation for Identification of Potential Poly (ADP-Ribose) Polymerase-1 (PARP-1) Inhibitors"

_molecules, 2019, doi:10.3390/molecules24234258_

Round 1
Reviewer 1 Report
Zhou et al. present a drug discovery study that evaluates the a novel screening method for the identification of potent and selective PARP-1 inhibitors. Although many PARP-1 inhibitors currently exist (most experimental, but some are in clinical trials and some are currently utilized in cancer treatment protocols), there is still a need due to adverse effects, new indications, and increased selectivity. The authors report the discovery of (potentially) such an inhibitor: one with potency and selectivity. The screening and discovery were rational and impressive, albeit a bit smoother than most likely occurred, and probably could be used for other PARP family member or even poly(ADP-ribose) glycohydrolase (PARG) in the future. Thus, the study is suitable for publication in Molecules with minor editing and inclusions. The results are significant to the PARP and cancer fields.
Comments:
Title needs re-wording; maybe add “inhibitors” to the end? Abstract: second sentence grammar incorrect; also, for 4th sentence, which database? Good introduction, showing the need for more specific PARP-1 inhibitors Results 2.1: which domain in PARP-1 do these features bind? Results Table 4: can you include similar data on Compound 4?
Many of these critiques are minor, but the study could be enhanced by addressing 4 and 5 above.
Author Response
Response to Reviewer 1 Comments
Point 1: English language and style
( ) Extensive editing of English language and style required
( ) Moderate English changes required
(x) English language and style are fine/minor spell check required
( ) I don't feel qualified to judge about the English language and style
Response 1: Thank you for the suggestion. We have carefully corrected some grammatical errors in the revised manuscript.
Point 2: Comments:
Title needs re-wording; maybe add “inhibitors” to the end?
Response 2: Thank you for the suggestion. The word “inhibitors” has been added in the revised manuscript (please see “Title”).
Point 3: Abstract: second sentence grammar incorrect;
Response 3: Thank you for the suggestion. The second sentence grammar error has been corrected in the revised manuscript (please see “Abstract: second sentence”).
Point 4: Comments:
for 4th sentence, which database?
Response 4: Thank you for the suggestion. The name of database in the 4th sentence has been added in the revised manuscript (please see “Abstract: 4th sentence”).
Point 5: Good introduction, showing the need for more specific PARP-1 inhibitors Results 2.1: which domain in PARP-1 do these features bind?
Response 5: Thank you for the suggestion. The name of the domain, “PARP-1 catalytic domain”, has been added in the revised manuscript (please see “Results 2.1”).
Point 6: Results Table 4: can you include similar data on Compound 4? Many of these critiques are minor, but the study could be enhanced by addressing 4 and 5 above.
Response 6: Thank you for the suggestion. The similar data on compound 4 has been added in the revised manuscript (please see “Results Table 4”).
Reviewer 2 Report
The authors report an integrated protocol that combines pharmacophore mapping, virtual screening and molecular docking used to identify Poly(ADP-ribose)polymerase-1 (PARP-1) inhibitors. 4 compounds were selected and confirmed through biological evaluation, confirming the results and validating the protocol and the conclusions.
The article is well written and the conclusions are well suported by the results. The protocol is well designed.
The following minor issues were noticed and should be taken into account:
The word inhibitors seems to be missing in the title at the end; The parameters of the VS total hits (Ht), active hits (Ha), enrichment factor (E), and goodness of hit score (GH) should be defined; section 3.5 which should explain VS protocol in reality is mainly used to describe the protocol used to convert and minimize the structures of the ligands when moving from 2D to 3D. Altough important this is described with too much detail, in contrast with the atual VS protocol, for which information is scare e and does not enable repetition Table 2, docking score, too many decimal number are presented. Values should be presented with 1 decimenal number (e.g. -13.9 kcal/mol)
Author Response
Response to Reviewer 2 Comments
Point 1: English language and style
( ) Extensive editing of English language and style required
( ) Moderate English changes required
(x) English language and style are fine/minor spell check required
( ) I don't feel qualified to judge about the English language and style
Response 1: Thank you for the suggestion. We have carefully corrected some grammatical errors in the revised manuscript.
Point 2: The following minor issues were noticed and should be taken into account:
The word inhibitors seems to be missing in the title at the end;
Response 2: Thank you for the suggestion. The word “inhibitors” has been added in the revised manuscript (please see “Title”).
Point 3:
The parameters of the VS total hits (Ht), active hits (Ha), enrichment factor (E), and goodness of hit score (GH) should be defined;
Response 3: Thank you for the suggestion. The parameters of the VS total hits (Ht), active hits (Ha), enrichment factor (E), and goodness of hit score (GH) have been defined in the revised manuscript (please see “Materials and methods: section 3.1”).
Point 4:
Section 3.2 which should explain VS protocol in reality is mainly used to describe the protocol used to convert and minimize the structures of the ligands when moving from 2D to 3D. Although important this is described with too much detail, in contrast with the atual VS protocol, for which information is scare e and does not enable repetition Table 2.
Response 4: Thank you for the thoughtful suggestion. The conformation import protocol available in MOE is used to convert and minimize the structures of the compounds using the MMFF94x force field when moving from 2D to 3D structures. In the process, multiple conformations per compound were generated and minimized until a root mean square gradient of 0.01 kcal mol-1 was obtained, the hydrogens are added and partial charges computed. The repetition has been removed. The related information has been added in the revised manuscript (please see “Materials and methods: section 3.2”).
Point 5:
Docking score, too many decimal number are presented. Values should be presented with 1 decimenal number (e.g. -13.9 kcal/mol)
Response 5: Thank you for the thoughtful suggestion. The values have been presented with 1 decimenal number in the revised manuscript (please see “Table 3”).
Reviewer 3 Report
The authors describe a screening protocol for identifying small molecule binders of the enzyme PARP-1 using a combination of pharmacophore modeling, virtual screening and experimental testing. Overall, the article is clearly written, although information is missing in certain areas. The method described is not novel, since pharmacophore based screening is an established protocol in the literature. However, their results, including four positive compounds are of reasonable significance in the therapeutic field. Here are a few comments/ criticisms:
The introduction doesn’t discuss the benefit of pharmacophore based pre-screening, as opposed to straight docking of the entire 35,000 ligand set. Also missing are the discussion of previous works where such pharmacophore based procedure was successfully used to detect positive hits in other targets. the authors make a case of finding novel compounds, but didn’t compare the structures of the four compounds with existing PARP inhibitors. Also, it would be helpful to overlay the binding poses of the compounds with NAD+, showing how the new compounds compete with NAD+. The authors describe their 35,000 compounds database as in-house, but also mentions ‘specs’. The specs databases offer hundreds of thousands of compounds. It should be clearly mentioned which specific subset database from specs was used, and how it was further curated, even if this was published before. It appears, For matching with the 3D pharmacophore model, only one structure per compound was used, which could be a hit or miss, especially for compounds with multiple rotatable bonds. Using multiple conformations per compound will have more confidence in pharmacophore matching. The fact that all four selected compounds were positive hits could have been a matter of luck. what cutoff in docking score was used to select the four compounds? Why were only the top 4 compounds experimentally tested instead of all 41, since 41 is not a large number.? By only testing four compounds, are the authors not taking risk, considering that virtual screening is still an imperfect science. As stated before, the success of all four hits could have been lucky, which works for this particular case. However, for generalizing this protocol to other targets, as the authors have proposed in the conclusion, they need to justify why their followed steps are reasonable. The experimental concentration dependent binding curves for all the compounds should be provided, along with error bars. Also, the IC50 values in all the tables should have error bars on them. I think, this work would have been more convincing if some in-vivo (e.g. mouse) studies accompanied the in-vitro testing to show the ultimate effectiveness of the compounds. In section 3.3 of methods, the authors write “binding free energy between tubulin and a ligand”. Should be PARP, not tubulin.
Author Response
Response to Reviewer 3 Comments
Point 1: English language and style
( ) Extensive editing of English language and style required
( ) Moderate English changes required
(x) English language and style are fine/minor spell check required
( ) I don't feel qualified to judge about the English language and style
Response 1: Thank you for the suggestion. We have carefully corrected some grammatical errors in the revised manuscript.
Point 2: Here are a few comments/ criticisms:
The introduction doesn’t discuss the benefit of pharmacophore based pre-screening, as opposed to straight docking of the entire 35,000 ligand set.
Response 2: Thank you for the suggestion. The benefit of pharmacophore based pre-screening has been added in the revised manuscript (please see “1. Introduction”).
Point 3: Also missing are the discussion of previous works where such pharmacophore based procedure was successfully used to detect positive hits in other targets.
Response 3: Thank you for the suggestion. In previous works, such pharmacophore-based procedure was successfully used to detect some positive hits in other targets such as histone methyltransferase 7 (SET7), soluble epoxide hydrolase (sEH) and enhancer of zeste homolog 2 (EZH2) [27-29]. The discussion of previous works has been added in the revised manuscript (please see “1. Introduction”).
Point 4: the authors make a case of finding novel compounds, but didn’t compare the structures of the four compounds with existing PARP inhibitors. Also, it would be helpful to overlay the binding poses of the compounds with NAD+, showing how the new compounds compete with NAD+.
Response 4: Thank you for the suggestion. The comparation of the structures of the four compounds with existing PARP inhibitors such as olaparib and niraparib (Figure S1) has been added in the revised manuscript (please see “1. Introduction and Figure S1”).
Point 5: The authors describe their 35,000 compounds database as in-house, but also mentions ‘specs’. The specs databases offer hundreds of thousands of compounds. It should be clearly mentioned which specific subset database from specs was used, and how it was further curated, even if this was published before.
Response 5: Thank you for the suggestion. The related spelling errors have been corrected in the revised manuscript (please see “3.2. Virtual screening”).
Point 6: It appears, for matching with the 3D pharmacophore model, only one structure per compound was used, which could be a hit or miss, especially for compounds with multiple rotatable bonds. Using multiple conformations per compound will have more confidence in pharmacophore matching.
Response 6: Thank you for the suggestion. The related information has been added in the revised manuscript (please see “3.2. Virtual screening”).
Point 7: The fact that all four selected compounds were positive hits could have been a matter of luck. what cutoff in docking score was used to select the four compounds? Why were only the top 4 compounds experimentally tested instead of all 41, since 41 is not a large number? By only testing four compounds, are the authors not taking risk, considering that virtual screening is still an imperfect science. As stated before, the success of all four hits could have been lucky, which works for this particular case. However, for generalizing this protocol to other targets, as the authors have proposed in the conclusion, they need to justify why their followed steps are reasonable.
Response 7: Thank you for the suggestion. Hit compounds identified by the pharmacophore model can be ranked according to the RMSD values, which is the degree of consistency with the pharmacophore model. Based on a RMSD value less than 0.5 Å, 41 selected compounds were docked into the PARP-1 active site. The docking scores between PARP-1 and 41 compounds were calculated by dG docking scoring function of MOE (lower docking scores indicate better binding affinity). Considering a cutoff to classify compounds as active and inactives, we used a -12 kcal mol-1 cutoff in docking score to prune the hit list. Finally, only 4 out of 41 compounds (compounds 1-4) below -12 kcal mol-1 were selected for further biological evaluation (Figure 3). The related information has been added in the revised manuscript (please see “2.2. Validation and database screening”).
Point 8: The experimental concentration dependent binding curves for all the compounds should be provided, along with error bars. Also, the IC50 values in all the tables should have error bars on them. I think, this work would have been more convincing if some in-vivo (e.g. mouse) studies accompanied the in-vitro testing to show the ultimate effectiveness of the compounds.
Response 8: Thank you for the suggestion. The binding curves for all the compounds have been added in the revised manuscript (please see “Figure S2”). The IC50 values with error bars in all the tables have been added in the revised manuscript (please see “Table 3”). To further evaluate the anticancer activity of all 4 compounds, MTT assay was used to test their inhibition to A549 cells. As shown in Figure 5, compounds 1-4 inhibited the growth of human lung cancer A549 cells in a dose-dependent manner. The related information has been added in the revised manuscript (please see “Figure 5”).
Point 9: In section 3.3 of methods, the authors write “binding free energy between tubulin and a ligand”. Should be PARP, not tubulin.
Response 9: Thank you for the suggestion. The related spelling errors have been corrected in the revised manuscript (please see “section 3.3”).
Reviewer 4 Report
The manuscript is well written, although the poor presentation. However, the authors have to incorporate the figures from the biological assays. Actually they use ONLY one table to present their data. Where are the figures from the binding assays? Usually when we have to present IC50 values we have to include the curves with the necessary data. They also dont includ any data for enzyme kinetics. In the Results & Discussion, they have to go in depth their thought.. they didnt mention anything about the perspectives and future plans for this work. As a result the conclusion paragraph is just a few lines.
Author Response
Response to Reviewer 4 Comments
Point 1: English language and style
( ) Extensive editing of English language and style required
( ) Moderate English changes required
(x) English language and style are fine/minor spell check required
( ) I don't feel qualified to judge about the English language and style
Response 1: Thank you for the suggestion. We have carefully corrected some grammatical errors in the revised manuscript.
Point 2: Comments and Suggestions for Authors
The manuscript is well written, although the poor presentation. However, the authors have to incorporate the figures from the biological assays. Actually they use ONLY one table to present their data. Where are the figures from the binding assays? Usually when we have to present IC50 values we have to include the curves with the necessary data. They also dont include any data for enzyme kinetics.
Response 2: Thank you for the suggestion. Figure S2 describes PARP-1 inhibition curves of compounds 1-4 and NU1025 in PARP-1 enzyme assay. The figure S2 has been added in supporting information (please see “Figure S2”). Figure 5 describes the inhibitory effects of all 4 compounds on A549 cell proliferation. The figure 5 from the cell assays has been added in the revised manuscript (please see “Figure 5”).
Point 3: In the Results & Discussion, they have to go in depth their thought.. they didnt mention anything about the perspectives and future plans for this work. As a result the conclusion paragraph is just a few lines.
Response 3: Thank you for the thoughtful suggestion. The related perspectives and future plans have been added in the revised manuscript (please see “Conclusion”).
Round 2
Reviewer 3 Report
In this version, the authors have addressed all the previous comments and queries. However, Fig. 5 raises new questions. Why are such high concentrations > 15 μM needed for killing the A549 cells, when the IC50s of the compounds are between 0.03 - 0.2 μM. The authors have used such high concentrations, that the question arises whether the tumor killing is due to the specific effect of interfering with the PARP-1 pathway, or more non-specific toxic effects.
Author Response
Point 1: English language and style
( ) Extensive editing of English language and style required
( ) Moderate English changes required
(x) English language and style are fine/minor spell check required
( ) I don't feel qualified to judge about the English language and style
Response 1: Thank you for the suggestion. We have carefully corrected some grammatical errors in the revised manuscript.
Point 2: Comments:
In this version, the authors have addressed all the previous comments and queries. However, Fig. 5 raises new questions. Why are such high concentrations > 15 μM needed for killing the A549 cells, when the IC50s of the compounds are between 0.03 - 0.2 μM. The authors have used such high concentrations, that the question arises whether the tumor killing is due to the specific effect of interfering with the PARP-1 pathway, or more non-specific toxic effects.
Response 2: Thank you for the suggestion. Presently, anticancer IC50 values estimated for most PARP-1 inhibitors range from 10 to 50 μM. For example, olaparib is an oral, small molecule, poly (ADP-ribose) polymerase inhibitor being developed by AstraZeneca for the treatment of solid tumors. In 2014, the drug olaparib became the first PARPi to receive FDA approval. In a previous study, PARP-1 enzyme IC50 value estimated for olaparib is 0.0155 μM and olaparib inhibited proliferation of A549 cells with the IC50 of 28 ± 3.4 μM (J Med Chem. 2014, 57: 2292–2302). In addition, olaparib inhibited proliferation of a dozen different types of non-small-cell lung cancer (NSCLC) cell lines with IC50 concentrations ranging from 10 to 50 μM concentrations (J Med Chem. 2014, 57: 2292–2302).
Reviewer 4 Report
Thank you for your reply.
My comments and suggestions are in a fruitful approach to increase the visibility and the citability of your paper
Author Response
Point 1: English language and style
( ) Extensive editing of English language and style required
( ) Moderate English changes required
(x) English language and style are fine/minor spell check required
( ) I don't feel qualified to judge about the English language and style
Response 1: Thank you for the suggestion. We have carefully corrected some grammatical errors in the revised manuscript.
Point 2: Thank you for your reply.
My comments and suggestions are in a fruitful approach to increase the visibility and the citability of your paper.
Response 2: Thank you very much for your attention and kindly advice! This is particularly important for increasing the visibility and the citability of our paper.